# Growth Performance and Immunity of Broilers Fed Sorghum–Soybean Meal Diets Supplemented with Phytases and Β-Mannanases

**DOI:** 10.3390/ani14060924

**Published:** 2024-03-17

**Authors:** Nicolás Sastré-Calderón, Gabriela Gómez-Verduzco, Arturo Cortés-Cuevas, Mireya Juárez-Ramírez, José Arce-Menocal, Claudia Cecilia Márquez-Mota, Félix Sánchez-Godoy, Ernesto Ávila-González

**Affiliations:** 1Departamento de Medicina y Zootecnia de Aves, Facultad de Medicina Veterinaria y Zootecnia, Universidad Nacional Autónoma de México, Avenida Universidad 3000, Ciudad de Mexico 04510, Mexico; mvz.nico.sastre@gmail.com (N.S.-C.); spuma92@hotmail.com (F.S.-G.); 2Centro de Enseñanza, Investigación y Extensión en Producción Avícola CEIEPAv, Tláhuac 13300, Mexico; cuevasarturo03@yahoo.com (A.C.-C.); avilaernesto@yahoo.com (E.Á.-G.); 3Departamento de Patología, Facultad de Medicina Veterinaria y Zootecnia, Universidad Nacional Autónoma de México, Avenida Universidad 3000, Ciudad de Mexico 04510, Mexico; mireyajr@fmvz.unam.mx; 4Facultad de Medicina Veterinaria y Zootecnia, Universidad Michoacana de San Nicolás de Hidalgo, Morelia 58000, Mexico; josearce_55@yahoo.com.mx; 5Departamento de Nutrición Animal y Bioquímica, Facultad de Medicina Veterinaria y Zootecnia, Universidad Nacional Autónoma de México, Avenida Universidad 3000, Ciudad de Mexico 04510, Mexico; c.marquez@unam.mx

**Keywords:** exogenous enzymes, broilers, gut, sorghum–soybean meal diet

## Abstract

**Simple Summary:**

Improving digestion in broilers in a natural manner benefits their health. The addition of exogenous enzymes to feed improves digestion and intestinal health. The use of these enzymes improved the weight of broiler chicks in a healthy and natural way.

**Abstract:**

Most grains and vegetable feedstuffs used in commercial poultry feed contain phytates and polysaccharides—non-starchy chemical structures that are not degraded by digestive tract enzymes. Exogenous enzymes optimize the use of dietary ingredients. This study aimed to determine whether combining β-mannanases (400 g/ton) and phytases in broiler sorghum–soybean diets could improve performance and immunity in broilers. Four diets were randomized in a 2 × 2 factorial design, with two phytase levels (500 or 1500 FTU/kg) and β-mannanase supplementation (0–400 g/ton; 158 million units/kg minimum enzyme activity). Six replicate battery cages of 10 chicks were fed each diet ad libitum. To assess cellular and humoral immune responses, 10 birds per treatment were euthanized on day 21. Supplementation with β-mannanase enzymes led to increased body weight and a higher feed conversion index (FCI) (*p* < 0.05). The phytase factor improved the FCI at 1500 FTU/kg (*p* < 0.05). Supplementation with β-mannanases improved the immune response by increasing the IgA concentration in the duodenum (95%) and total serum immunoglobulins (*p* < 0.05). The morphometric index increased in all organs (*p* < 0.05), and the heterophile/lymphocyte ratio (HLR) decreased by 50% (*p* < 0.05). Supplementing broilers with β-mannanases in sorghum–soybean meal diets with phytases improved their performance and immunity.

## 1. Introduction

Enzymes have been developed as feed additives to improve the digestion and absorption of nutrients. The first phytase products entered the market in 1991 and have since been used extensively in monogastrics. Supplementation with exogenous phytases in cereal vegetable-based diets has been shown to improve monogastric animal production and can also contribute to breaking down phytates. This reduces phosphorus contamination and improves the utilization of phytic phosphorus and myo-inositol [1,2]. The recommended phytase doses in poultry farming range between 500 and 1000 FTU/kg [3,4]. A phytase super-dose is an amount higher than the recommended dose. Phosphorus is considered an essential inorganic mineral and incurs a high cost in diet rations [5]. Phytase supplementation could reduce phosphate excretion into the environment through animal waste.

On the other hand, it has been reported that supplementation with mannan-oligosaccharides (MOS) modulates the immune response [6]. Mannan-oligosaccharides are derived from the outer membranes of the cell walls of bacteria, plants, or yeasts [7]. Mannan-oligosaccharides have also been shown to suppress bacterial proliferation and favor the development of intestinal tract microbiota [8,9]. β-mannanases are endohydroxylases that hydrolyze β-D-1,4 mannopyranosid bonds, generating MOS such as mannobiose and mannotriose [10]. When β-mannans are hydrolyzed, metabolizable energy (ME) and other nutrients are also released, and the viscosity of the intestinal contents is reduced [11]. In the search for natural alternatives to improve performance in poultry, the aim of this research was to determine whether there was an interaction between β-mannanases (0–400 g/ton) and phytases (500 FTU/kg or 1500 FTU/kg) in broilers fed sorghum–soybean diets that could improve performance, local and systemic immunity, and intestinal integrity.

## 2. Materials and Methods

All the procedures involving broiler chickens used in this study were approved by the Institutional Animal Care and Use Committee (Comité Institucional para el cuidado y uso de los animales experimentales-CICUAE FMVZUNAM) and conducted according to Official Mexican Norm (NOM-033-SAG/ZOO-2014) guidelines for animal welfare and experimental protocols.

### 2.1. Bird Husbandry

A total of 240 one-day-old Ross × Ross 308 broiler chicks of both sexes (50:50 ratio) (Aviagen North America, Huntsville, AL, USA) were obtained from a commercial hatchery and randomly distributed. The broiler chicks were raised in wire-floored cages (37.3 cm × 49.5 cm × 24.13 cm, Petersime^®^ Battery Cages (Petersime Incubator Co., Gettysburg, OH, USA)). Each cage was equipped with one feeder and one drinker. The birds had ad libitum access to water and mash feeds. The temperature during the experiment was lowered gradually from 32 °C at 1 day of age to 21 °C at 21 days of age. The light cycle was as follows: 13 h ± 20 min naturally L (light) and 11 h ± 20 min D (dark) per day.

### 2.2. Experimental Design and Diets

The broiler chickens were randomly allotted to one of four treatments in a 2 × 2 factorial arrangement. The basal diets comprised sorghum and soybean meals, with phytase supplementation (Axtra^®^ PHY; Danisco Animal Nutrition PO Box 777, Marlborough, Wilts SN8 1XN, UK, 500 FTU/kg or 1500 FTU/kg) and Hemicell^®^ (HT β-mannanases) (158 million units/kg minimum enzyme activity, Elanco Animal Health, 2500 Innovation Way Greenfield, IN 46,140 USA, 0 and 400 g/ton). The compositions of the experimental diets are provided in Table 1. The diets were designed according to the recommendations and guidelines in the Ross Nutrition Specifications 2022 [12]. The treatments were as follows: Treatment 1—sorghum–soybean meal basal diet with 500 FTU/kg of Axtra^®^ phytases; Treatment 2—sorghum–soybean meal diet + 1500 FTU/kg of Axtra^®^ phytases; Treatment 3—sorghum–soybean meal basal diet with + 500 FTU/kg of Axtra^®^ phytases + 400 g/ton of Hemicell^®^ β-mannanases; Treatment 4—sorghum–soybean meal basal diet with + 1500 FTU/kg of Axtra^®^ phytases + 400 g/ton of Hemicell^®^ β-mannanases.

### 2.3. Productive Performance

During the experiment, the broilers were weighed weekly to determine weight gain, feed intake, the coefficient of variation (Standard Deviation/Mean × 100), and the feed conversion index. The feed conversion index was the total amount of feed consumed by the flock divided by the amount of weight gained until 21 days of age.

### 2.4. Systemic Humoral Immune Response

To evaluate the systemic humoral immune response, on the 10th day, the broilers were simultaneously vaccinated with a live-virus vaccine against Newcastle disease via the ocular route, and a dead virus vaccine against Newcastle disease subcutaneously (Laboratorios Avilab, S.A. de C.V. Mexico; Tepatitlán de Morelos, Jal. La Sota^®^ Newcastle strain and Newcastle Plus^®^, respectively). On day 21, 2 mL of blood was taken from ten chicks from each treatment, and sera were obtained and frozen at −20 °C to determine the serum titers of antibodies specific to the ND virus through the hemagglutination inhibition test [13].

### 2.5. Quantification of Intestinal Immunoglobulin A (IgA) Antibodies

To calculate the total (unspecific) local IgA production in the epithelia of the gut, we utilized the ELISA chicken IgA quantitation kit (Bethyl Laboratories, Inc., Montgomery, TX, USA), following the manufacturer’s recommendations, at 21 days of age. Ten broilers per treatment were euthanized by cervical dislocation, and a 10 cm section was removed from the duodenum of each broiler [14].

### 2.6. Cellular Immune Response

Blood samples were collected with an ethylenediaminetetraacetic acid (EDTA) S-Monovette (Sarstedt AG & Co.KG, Sarstedtstraße 1, 51,588 Nümbrecht, Alemania) from the radial veins of 21-day-old broilers (10 chickens per treatment). A differential leukocyte count was performed using blood smears stained with Wright’s stain. Total counts were indirectly determined by calculating the cell and total count percentages [15].

### 2.7. Relative Weight of the Organs

The spleens, Fabricius bursae, and guts of ten broilers per treatment were weighed. The weight of each organ was divided by the body weight of each broiler per 100 as previously reported [16,17].

### 2.8. Statistical Analysis

The results obtained for the variables under study were analyzed to verify compliance with the assumptions of normality of the residuals and homoscedasticity of variances, with a significance level of *p* < 0.05, for parametric analyses of the data according to a completely randomized design with a 2 × 2 factorial arrangement. The first factor corresponded to the two phytase levels (500 and 1500 FTU/kg), whereas the second factor was 0 or 400 g/ton β-mannanases. The data for the antibody titers vs. Newcastle disease were transformed to logarithm base 2. The results for the evaluated variables were analyzed using JMP^®^ computer software, version 8.

## 3. Results

### 3.1. Animal Performance

The results for the productive variables are shown in Table 2. Broilers fed with the β-mannanase enzymes showed an improvement in productive performance (*p* < 0.05), weight (940.7 vs. 863.5), FCI (1.12 vs. 1.19), and CV (4.57 vs. 7.77). Phytase affected the FCI (*p* < 0.05), and its beneficial effect was 2 percentage points lower when a dose of 1500 FTU/kg was used (1.14 vs. 1.16). No differences in the feed consumption and body weight for other phytase doses were observed. No interaction effect between the phytase levels or the addition of β-mannanases and improved productive variables was observed. (*p* > 0.01).

### 3.2. Systemic and Local Humoral Immune Response

The results for the immunological variables are shown in Table 3. Diets containing β-mannanases showed a difference at *p* < 0.05, resulting in a higher systemic serum concentration of immunoglobulins against Newcastle virus (6.00 vs. 5.25). Additionally, a positive effect was observed for the total intestinal IgA concentration; this concentration was higher following the treatments containing β-mannanases [95% (523.46 vs. 267.99)] than following the consumption of diets without β-mannanases. There was no interaction effect (*p* > 0.05) between the factors on serum ND antibody titers.

### 3.3. Hematology

No effects were observed for any of the factors or interactions for the following variables in the blood (*p* > 0.01), leukocytes, eosinophils, basophils, and monocytes (data not included). The results for the heterophile/lymphocyte ratio (H/L) are shown in Table 4. The cellular profile showed significant differences in heterophiles, lymphocytes, and H/L ratio (*p* > 0.01). Diets containing β-mannanases resulted in lower heterophiles (2.67 vs. 4.20) and a higher number of lymphocytes (10.66 vs. 7.89), and the HLR was reduced by 55% (0.25 vs. 0.56). Phytase did not result in any observable statistically significant difference in the cellular profile, nor was any interaction effect between factors observed.

### 3.4. Relative Organ Weights

The relative gut weight (RGW), Fabricius bursa weight (RBW), and spleen weight (RSW) are shown in Table 5. No interaction effect between factors was observed. However, supplementation with β-mannanases (400 g/ton) resulted in differences in all treatments for the RGW (5.81 vs. 6.28), RSW (0.11 vs. 0.07), and RBW (0.22 vs. 0.18) compared with the phytase groups and β-mannanases at 0 g/ton. Phytase supplementation did not result in any observable statistically significant differences in the evaluated variables.

## 4. Discussion

### 4.1. Animal Performance

In the present study, the impact of different doses of phytase and β-mannanases on broiler growth performance, feed conversion efficiency, systemic, local humoral response, and heterophile/lymphocyte ratio was investigated. The results indicate that there was no significant difference in outcomes between phytase doses of 500 and 1500 FTU (*p* > 0.05). This suggests that the lower dose of phytase was sufficient in meeting the phosphorus requirements of the broilers, indicating that the diet composition with the lower phytase dose was adequate [18].

The improvement in daily weight gain achieved with β-mannanases could be attributed to the fact that enzymes in sorghum–soybean meal diets act against β-mannans present in the cell structure of the soybean meal, an ingredient widely used in broilers’ diets. β-mannans are non-starchy polysaccharides and function as an antinutritional factor; therefore, their degradation prevents a negative effect on intestinal viscosity and increases nutrient absorption [19]. In addition, they eliminate the energy expenditure that results from the immune response induced by food, and they improve energy availability due to the breakdown of compounds.

Previous studies [20,21,22,23] obtained results consistent with those obtained in this study in terms of growth improvement; in this study, an 8.9% improvement was found. β-mannanases also improved the FCI (6.2%); this finding was consistent with the findings of several studies [20,22,23]. A 0.067 g reduction was previously reported [20], similar to that in our study, where a decrease of 0.07 g was observed.

Moreover, the utilization of β-mannanases not only enhances the growth performance but also diminishes the energy expenditure connected with the immune response generated by antinutritional elements such as β-mannans. By decomposing these compounds, β-mannanases augment the accessibility of energy in the diet, thereby contributing to improved overall performance in broilers. The present study enhances the significance of supplementing enzymes, particularly β-mannanases, in ameliorating the growth performance and feed efficiency of broilers by targeting antinutritional elements like β-mannans in the diet. The outcomes are consistent with previous research, further validating the effectiveness of β-mannanases in enhancing broiler production.

### 4.2. Systemic and Local Humoral Immune Response

The evaluation of the impact of these molecules on the immune system showed that broilers fed with β-mannanase-containing diets had higher titers of antibodies against the Newcastle disease virus and a higher concentration of secreted IgA in the duodenum, and the weights and morphometric indices of their lymphoid organs were higher. This response may be attributed to the molecular mechanisms of β-mannanase, which degrades β-mannans into MOS, which are known to be immunostimulants. Due to stimulation of the lymphoid tissue associated with the intestine, leading to MOS being recognized as pathogen-associated molecular patterns, the immune response is stimulated [24]. Mannose receptors also become occupied by MOS. The combination of both mechanisms triggers a better immune response. In this study, this effect was verified by the increase in IgA secretions [25].

The use of β-mannanases in the diet increased the morphometric indices of the spleen and Fabricius bursa. Larger spleens and Fabricius bursae [26] have been correlated with a higher IgY concentration, according to previous studies [27] in which higher titers of antibodies were found when β-mannanases were included in the diet.

### 4.3. Hematology

The heterophile/lymphocyte ratio showed a 55% reduction when β-mannanases were provided compared to when the diet lacked the enzyme, as previously reported [28,29]. A reduction in this parameter has been associated with increased immunity, which is likely attributable to the reduction in stress among the broilers.

Recently, it was determined that MOS do not function as typical prebiotics, which, by definition, are non-digestible ingredients in the diet that stimulate the growth and/or activity of beneficial bacteria in the digestive tract, promoting general and intestinal health [30,31]. We found that including β-mannanases in the diet resulted in differences in intestinal weight. No studies have previously reported the effects of β-mannanases on these parameters. Intestinal weight is not related to nutrient digestibility [32]. Based on these data, we can speculate that the improvement in relative gut weight was caused by a decrease in viscosity in the intestinal lumen due to β-mannans. Alternatively, an increase in the mass of intestinal microbiota has been proposed as a mediating factor.

## 5. Conclusions

This study found that supplementing broiler diets containing sorghum and soybean meals with β-mannanases and phytases can improve the performance, immune function, and intestinal integrity of broilers. These findings have practical implications for optimizing commercial poultry diet formulations, highlighting the importance of supplementation with enzymes in unlocking the full nutritional potential of grains and vegetable feedstuffs. Further research and the application of these insights in the poultry industry may result in more efficient and sustainable practices, benefiting both producers and consumers.

## Figures and Tables

**Table 1 animals-14-00924-t001:** Composition and calculated analysis of the experimental starting basal diet for broilers (1–21 days of age).

Ingredients	Basal Diet
Sorghum	571.4
Soybean Meal	370.9
Vegetable Oil	18.4
Calcium Carbonate	14.5
Orthophosphate	10.1
Salt	3.5
DL-Methionine	3.1
Vitamin Premix *	3
Mineral Premix **	0.5
L-Lysine HCl	2.8
L-Threonine	0.7
IQ (Antioxidant) ***	0.15
Cellulose	0.45
Axtra^®^ PHY	0.1
Choline Chloride 60%	0.05
Total	1000
Calculated Analysis	
Metabolizable Energy (Kcal/kg)	3010
Crude Protein (%)	22
Digestible Lysine (%)	1.44
Digestible Met + Cis (%)	0.9
Total Arginine (%)	1.3
Sodium (%)	0.18
Total Calcium (%)	0.96
Available Phosphorus (%)	0.48

* Commercial nucleus contribution: Vit. A 12,000,000 UI; Vit. D3 2,500,000 UIP; Vit. E 15,000 UI; Vit. K3 2000 mg/kg; Vit. B1 2250 mg/kg; Vit. B2 7500 mg/kg; Vit. B3 45,000 mg/kg; Vit. B5 12,500 mg/kg; Vit. B6 3500 mg/kg; Vit. B12 20 mg/kg; Folic Acid 1500 mg/kg; Biotin 125 mg/kg. ** For 0.5 kg: Iodine 300 mg/kg; Selenium 200 mg/kg; Cobalt 200 mg/kg; Iron 50,000 mg/kg; Copper 12,000 mg/kg; Zinc 50,000 mg/kg; Manganese 110,000 mg/kg. *** BHT (1.2%) and BHQ (9%).

**Table 2 animals-14-00924-t002:** Interaction of phytase and β-mannanases in the productive responses of broilers at 21 days of age.

Phytase (FTU/kg)	β-Mannanases	Weight (g)	FC (g)	DWG (g)	FCI (kg)	CV (%)
500	400 g/ton	928	1006	42.1	1.12	4.78
	0	857	996	38.7	1.2	8.64
1500	400 g/ton	954	1017	43.3	1.11	4.36
	0	870	992	39.3	1.17	6.9
Phytase	500	892	1001	40.4	1.160 ^a^	6.71
	1500	912	1004	41.3	1.140 ^b^	5.63
β-mannanases	400 g/ton	940 ^a^	1011	42.7 ^a^	1.120 ^b^	4.57 ^b^
	0	863 ^b^	994	39.1 ^b^	1.190 ^a^	7.77 ^a^
*p*-value						
Phytase		0.175	0.89	0.188	0.032	0.133
β-mannanases		0.001	0.471	0.001	0.001	0.001
Phytase × β-mannanases		0.63	0.75	0.645	0.23	0.379
MSE		9.8	16.8	0.46	0.007	-

The absence of literals between the means of each column indicates that there were no statistically significant differences (*p* > 0.01). MSE = mean standard error; FC: feed consumption; DWG: daily weight gain; FCI: feed conversion index (total amount of feed consumed by the flock divided by the amount of weight gained); CV: coefficient of variation (Standard Deviation/Mean × 100).

**Table 3 animals-14-00924-t003:** Humoral immune response (local and systemic) in broilers fed with phytase and β-mannanases at 21 days of age.

Phytase (FTU/kg)	β-Mannanases	Intestinal IgA (ng/mL)	HI ND
		OD 405 nm	Log 2
500	400 g/ton	436.2	5.83
	0	228.2	5
1500	400 g/ton	610.7	6.17
	0	307.7	5.5
Phytase	500	332.2	5.42
	1500	459.2	5.83
β-mannanases	400 g/ton	523.5 ^a^	6.00 ^a^
	0	267.9 ^b^	5.25 ^b^
*p*-value			
Phytase		0.667	0.223
β-mannanases		0.002	0.035
Phytase × β-mannanases		0.123	0.804
MSE		55.5	0.23

The absence of literals between the means of each column indicates that there were no statistically significant differences (*p* > 0.05). HI = hemagglutination inhibition; ND = Newcastle disease; OD = optical density; MSE = mean standard error.

**Table 4 animals-14-00924-t004:** Analysis of immune system cells using a hemogram in broilers fed with sorghum + soybean meal diets containing phytase and β-mannanases at 1–21 days of age.

Phytase (FTU/kg)	β-Mannanases	Heterophiles (×10^9^/L)	Lymphocytes (×10^9^/L)	H/L Ratio
500	400 g/ton	2.38	10.2	0.23
	0	4.65	8.58	0.57
1500	400 g/ton	2.95	11.11	0.27
	0	3.75	7.2	0.54
Phytase	500	3.52	9.39	0.4
	1500	3.35	9.16	0.41
β-mannanases	400 g/ton	2.67 ^a^	10.66 ^a^	0.25 ^a^
	0	4.20 ^b^	7.89 ^b^	0.56 ^b^
*p*-value				
Phytase		0.117	0.277	0.724
β-mannanases		0.009	0.001	0.001
Phytase × β-mannanases		0.804	0.804	0.239
MSE		0.27	0.77	-

The absence of literals between the means of each column indicates that there were no statistically significant differences (*p* < 0.01). H/L ratio = heterophile/lymphocyte ratio, MSE = mean standard error.

**Table 5 animals-14-00924-t005:** Weights and morphometric indices of organs from broilers fed with sorghum + soybean meal diets containing phytase and β-mannanases at 21 days of age.

Phytase (FTU/kg)	β-Mannanases	GW (g)	RGW (%)	SW (g)	RSW (%)	FBW (g)	RFBW (%)
500	400 g/ton	53.67	5.84	0.98	0.11	1.9	0.21
	0	51.33	6.5	0.48	0.06	1.4	0.17
1500	400 g/ton	56.33	5.77	1.07	0.11	2.2	0.22
	0	52.83	6.05	0.68	0.08	1.67	0.19
Phytase	500	52.5	6.17	0.73	0.09	1.65	0.19
	1500	54.58	5.91	0.88	0.09	1.91	0.21
β-mannanases	400 g/ton	55.00 ^a^	5.81 ^a^	1.03 ^a^	0.11 ^a^	2.03 ^a^	0.22 ^a^
	0	52.08 ^b^	6.28 ^b^	0.58 ^b^	0.07 ^b^	1.54 ^b^	0.18 ^b^
*p*-value							
Phytase		0.075	0.404	0.059	0.116	0.057	0.234
β-mannanases		0.003	0.006	0.001	0.001	0.001	0.026
Phytase × β-mannanases		0.51	0.14	0.384	0.285	0.946	0.947
MSE		0.61	-	0.044	-	0.086	-

The absence of literals between the means of each column indicates that there were no statistically significant differences (*p* < 0.05). GW = gut weight; RGW = relative gut weight; SW = spleen weight; RSW = relative spleen weight; FBW = Fabricius bursa weight; RFBW = relative Fabricius bursa weight; MSE = mean standard error.

## Data Availability

The datasets generated during and/or analyzed during the study are available from the corresponding author upon reasonable request.

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
