# Peer review of "Growth Performance and Immunity of Broilers Fed Sorghum–Soybean Meal Diets Supplemented with Phytases and Β-Mannanases"

_animals, 2024, doi:10.3390/ani14060924_

Round 1

Reviewer 1 Report

Comments and Suggestions for Authors

The current study assessed the role of phytase and β-mannanase in improving the productive performance of broilers. The subject is important for broiler feeding. However, the novelty of the study is not addressed. The manuscript needs extensive English editing for grammar, typing mistakes, commas, and full stops. The writing style is bad.

L25: correct spelling of polysaccharides and non-amilaceous.

L29: 0 and 400.

L55: EM? Define

The introduction is weak, and the writing style is bad. The author should strengthen it with more recent literature and clarify the study's novelty.

L73: were?? Where is the object?

Table 1: cite the reference for diet formulation. What was the length of the experiment?

Why only the diet for the starter is provided? Were the enzymes tested for the starter period only? Why, if so?

L136: CV? DEFINE. Where are the calculations for the performance measures?

L136-141: in comparison with what? The results should be rewritten and clarified.

Table 2: revise the superscripts for FCI in “phytase”

The results of BWG are significant for β-mannanases. Add in the text.

L146-151: the results should be detailed and clarified.

HLR? Define. All abbreviations should be defined at first mention.

Table 4: revise the letters for higher and lower values. The same for table 5.

L174, 175: compared with??

L214: Heterofile:Lymphocite. Correct spelling

There is no discussion on the effect of phytase or its interaction with β-mannanases.

Comments on the Quality of English Language

Extensive editing is needed 

Author Response

Title

Growth Performance and Intestinal Health Of Broilers Fed Sorghum-Soybean Meal Diets Supplemented with Phytases and Β-Mannanases

REVIEWER 1

On were changed by of

Ln 25. Non- starchy observation made, thank you.

Line 29.  Were 0 includes. Request made, thank you

Ln 55 ME, made define ME Request made, thank you.

Ln 63-90. Al Materials and Methods were rewrite to properly state meaning.  Thank you, paragraph was revised and modified.

Line 73- Were and where is the object?. All material and methods were rewrite and Request made, thank you

Table 1: cite the reference for diet formulation. What was the length of the experiment? The diets were made according to the requirements stipulated in the Ross 308 strain manual. In the starter stage. Your observation  in the length of the experiment was already made when rewriting the material and methods.

Why only the diet for the starter is provided?. This experiment only focused on the initiation stage.  Were the enzymes tested for the starter period only? Why, if so? The starter stage is considered the most critical stage in the productive development of broilers; For example, in the first week it quadruples its initial weight; The aim was to evaluate whether increasing the dose of phytases   which is usually used from 500 to 1500, there was an effect on the productive performance on broilers.

L136: CV? DEFINE. is a statistical measure of the dispersion of data points around the mean. Where are the calculations for the performance measures? CV = standard deviation / sample mean x 100 . It was included in line 100 to 103. In the same way, it was placed in the footer of table 2

L136-141: in comparison with what? The results should be rewritten and clarified. suggestion has been made

Table 2: revise the superscripts for FCI in “phytase” suggestion has been made

The results of BWG are significant for β-mannanases. Add in the text. . suggestion has been made. Thank you .

L146-151: the results should be detailed and clarified. suggestion has been made. Thank you .

HLR? Define. All abbreviations should be defined at first mention. apologize, it was a typing error, it refers to the H/L ratio. Thank you

Table 4: revise the letters for higher and lower values. The same for table 5. suggestion has been made. Thank you .

L174, 175: compared with?? suggestion has been made. Thank you .

L214: Heterofile:Lymphocite. Correct spelling. . suggestion has been made. Thank you . 

There is no discussion on the effect of phytase or its interaction with β-mannanases. In the results tables shown, no significant interactions between treatments are evident. The autor do not considered to comment on the discussion about it. We thought that there could be some type of beneficial interaction; However, our statistical data did not show interactions

Reviewer 2 Report

Comments and Suggestions for Authors

LINE 48-49: Please, uniform the dimension of characters.

LINE 70 Experimental design: Please, add the day(light)/night(dark) cycle of the subjects of the experiment. The authors should explain why there isn’t a placebo control feed with only Sorghum soybean meal basal diet, this is a lack in the experimental design.

LINE 126 Statistical analysis: Statistical analysis and test carried out are not clear-sighted. Please specify the type of test used to face the different variables analysed among the different treatment group (distribution test, homoscedasticity test and subsequently parametric or not parametric test with a suitable post-hoc test to highlight the differences between the groups). The lack of the explained statistic techniques applied is heavy. Please, add it and specify it, conversely the outcomes won’t be clear. Without this information clearly specified, write if there is a statistically difference or not is not sufficient. Moreover, write the exact p-value obtained in statistic analysis in every point of the study (not p< or p> 0.05).

LINE 192: Report the references just at the end of the phrase.

I would like to give an advice to the authors: the paper could be improved with more analysis of the blood samples (SDS-PAGE for detecting differences in proteomic profiles, electrophoresis in agarose gel for detecting differences in alpha, beta and gamma zone, measuring the main biomarkers). 

Author Response

thank you for your valuable observations I rewrote several paragraphs that you request. If it is considere necessary, I will send it to the English edition.

REVIEWER 2

Ln 48-49 Please, uniform the dimension of characters. Request made, thank you

LINE 70 Experimental design: Please, add the day(light)/night(dark) cycle of the subjects of the experiment. The authors should explain why there isn’t a placebo control feed with only Sorghum soybean meal basal diet, this is a lack in the experimental design. The suggestion was made, the material and methods were rewritten in an appropriate manner. thank you

LINE 126 Statistical analysis: Statistical analysis and test carried out are not clear-sighted. Please specify the type of test used to face the different variables analysed among the different treatment group (distribution test, homoscedasticity test and subsequently parametric or not parametric test with a suitable post-hoc test to highlight the differences between the groups). The lack of the explained statistic techniques applied is heavy. Please, add it and specify it, conversely the outcomes won’t be clear. Without this information clearly specified, write if there is a statistically difference or not is not sufficient. Moreover, write the exact p-value obtained in statistic analysis in every point of the study (not p< or p> 0.05). . The excellent  suggestion was made, the statistical analysis were rewritten in an appropriate manner. thank you

LINE 192: Report the references just at the end of the phrase. . Request made, thank you it is 18 reference

I would like to give an advice to the authors: the paper could be improved with more analysis of the blood samples (SDS-PAGE for detecting differences in proteomic profiles, electrophoresis in agarose gel for detecting differences in alpha, beta and gamma zone, measuring the main biomarkers). Thank you very much for the excellent suggestion, we would really like to be able to work with this proposal; perhaps in later studies it would be very enriching. For now, the bachelor student who carried out this work has already obtained his academic degree.

Round 2

Reviewer 1 Report

Comments and Suggestions for Authors

The authors ignored the reviewer's comments. The manuscript still needs language editing

The authors didn’t respond to these comments:

Table 1: cite the reference for diet formulation. What was the length of the experiment?

Why only the diet for the starter is provided? Were the enzymes tested for the starter period only? Why, if so?

In line 80, the authors said that the chicks were reared till 28 days of age, while In line 102, they said that the experiment length was 21 days. Please clarify the experiment design, period, and feeding stages.

Where are the calculations for the performance measures?

What is the coefficient of variation (Standard Deviation / Mean * 100)? Mean of what? What is its relation with performance parameters?

The writing style of the results still has not improved.  

The superscripts in the means are still incorrect. Sometimes, the letter “a” expresses higher values and sometimes lower values.

Previous revision was not made.

L136-141: in comparison with what? The results should be rewritten and clarified.

Table 2: revise the superscripts for FCI in “phytase”

The results of BWG are significant for β-mannanases. Add in the text.

L146-151: the results should be detailed and clarified.

HLR? Define. All abbreviations should be defined at first mention.

The discussion still has not improved.

Comments on the Quality of English Language

Extensive editing is needed 

Author Response

01-03-2024

REVIEWER 1

Table 1: cite the reference for diet formulation. What was the length of the experiment?

Why only the diet for the starter is provided? Were the enzymes tested for the starter period only? Why, if so?

  1. This experiment only focused on the initiation stage. Were the enzymes tested for the starter period only? Why, if so? The starter stage is considered the most critical stage in the productive development of broilers; For example, in the first week it quadruples its initial weight; The aim was to evaluate whether increasing the dose of phytases which is usually used from 500 to 1500, there was an effect on the productive performance on broilers.

In line 80, the authors said that the chicks were reared till 28 days of age, while In line 102, they said that the experiment length was 21 days. Please clarify the experiment design, period, and feeding stages.

  1. Thank you for the observation the length of the experiments, the correct length is days. We corrected in line 80. About the period and feeding stages.

Where are the calculations for the performance measures?

  1. In line 101-103 we included the calculations of performance measures.

What is the coefficient of variation (Standard Deviation / Mean * 100)? Mean of what? What is its relation with performance parameters?

  1. We included the calculation of coefficient of variation to explain the dispersion of the data and therefore to explain the effect of the treatments with the performance parameters.

The writing style of the results still has not improved.  

  1. We send the paper to English editing to improve the writing style.

The superscripts in the means are still incorrect. Sometimes, the letter “a” expresses higher values and sometimes lower values.

  1. Excellent observation. Request have made
  2. Thank you for your comments, we apologize for the inconvenience. We mistakenly uploaded a different work in round one of revisions.

L136-141: in comparison with what? The results should be rewritten and clarified.

  1. We clarified the results section, we added the required information in lines 145-148.

Table 2: revise the superscripts for FCI in “phytase”

The results of BWG are significant for β-mannanases. Add in the text.

  1. We included the effect of β-mannanases on weight (line 141)

L146-151: the results should be detailed and clarified.

  1. We clarified the results section, we added the required information in lines 145-148.

HLR? Define. All abbreviations should be defined at first mention.

  1. HRL is define in line 36.

The discussion still has not improved.

  1. We improved the discussion section.

Reviewer 2 Report

Comments and Suggestions for Authors

L74-81: Why is the light cycle continuos, while conventionally the basal cycle of day light includes dark period? Write a light cycle of 24:0 L(light)/D(Dark) and motive why. Please, write "A total of 240... both sexed (50:50 ratio)"

Furthermore, provide the exact p-value of your statistic test.

The authors have made significant improves.

Author Response

REVIEWER 2

The authors of this article appreciate the time spent reviewing this paper.

R2's second round of comments:

L74-81: Why is the light cycle continuos, while conventionally the basal
cycle of day light includes dark period? Write a light cycle of 24:0
L(light)/D(Dark) and motive why.

Thank you for your excellent observation, the text was NOT clear; the term “ continuos” made using to the fact that “the light that chickens received was constant;  because it was always natural light at that time of year ( June). In line 80 we tryed to explain this process according to your observation. the latest changes are marked in green in text

Please, write "A total of 240... both sexed.              The request was made . Thank you
(50:50 ratio)"  the latest changes are marked in green in text

Furthermore, provide the exact p-value of your statistic test.

: Excellent observation, changes were made to the lines : Line 148, 150, 168,171 and also in  foot of the tables 2 and 4. Thank you  the request was made . the latest changes are marked in green in text

Round 3

Reviewer 1 Report

Comments and Suggestions for Authors

the authors should cite the reference of the diet in the text

Comments on the Quality of English Language

minor editing

Author Response

Dear reviewer The authors of this article are very grateful for your excellent suggestions. The request has already been made: on line 90 within the article text. Also
in the bibliographic reference was included, number 12